# Plasma Metabolite Profiling in the Search for Early-Stage Biomarkers for Lung Cancer: Some Important Breakthroughs

**DOI:** 10.3390/ijms25094690

**Published:** 2024-04-25

**Authors:** Jill Meynen, Peter Adriaensens, Maarten Criel, Evelyne Louis, Karolien Vanhove, Michiel Thomeer, Liesbet Mesotten, Elien Derveaux

**Affiliations:** 1Faculty of Medicine and Life Sciences, Hasselt University, Martelarenlaan 42, B-3500 Hasselt, Belgium; jill.meynen@uhasselt.be (J.M.); maarten.criel@zol.be (M.C.); karolien.vanhove@uzleuven.be (K.V.); liesbet.mesotten@zol.be (L.M.); 2Applied and Analytical Chemistry, NMR Group, Institute for Materials Research (Imo-Imomec), Hasselt University, Agoralaan 1, B-3590 Diepenbeek, Belgium; elien.derveaux@uhasselt.be; 3Department of Respiratory Medicine, Ziekenhuis Oost-Limburg, Synaps Park 1, B-3600 Genk, Belgium; michiel.thomeer@zol.be; 4Department of Respiratory Medicine, University Hospital Leuven, Herestraat 49, B-3000 Leuven, Belgium; evelyne.louis@uzleuven.be; 5Department of Respiratory Medicine, Algemeen Ziekenhuis Vesalius, Hazelereik 51, B-3700 Tongeren, Belgium; 6Department of Nuclear Medicine, Ziekenhuis Oost-Limburg, Synaps Park 1, B-3600 Genk, Belgium

**Keywords:** biomarkers, lung cancer, metabolomics, NMR (nuclear magnetic resonance)

## Abstract

Lung cancer is the leading cause of cancer-related mortality worldwide. In order to improve its overall survival, early diagnosis is required. Since current screening methods still face some pitfalls, such as high false positive rates for low-dose computed tomography, researchers are still looking for early biomarkers to complement existing screening techniques in order to provide a safe, faster, and more accurate diagnosis. Biomarkers are biological molecules found in body fluids, such as plasma, that can be used to diagnose a condition or disease. Metabolomics has already been shown to be a powerful tool in the search for cancer biomarkers since cancer cells are characterized by impaired metabolism, resulting in an adapted plasma metabolite profile. The metabolite profile can be determined using nuclear magnetic resonance, or NMR. Although metabolomics and NMR metabolite profiling of blood plasma are still under investigation, there is already evidence for its potential for early-stage lung cancer diagnosis, therapy response, and follow-up monitoring. This review highlights some key breakthroughs in this research field, where the most significant biomarkers will be discussed in relation to their metabolic pathways and in light of the altered cancer metabolism.

## 1. Introduction

Lung cancer is the leading cause of cancer-related mortality worldwide, with an overall five-year survival rate varying from 4 to 17% [1]. These low survival rates are mainly caused by delayed diagnosis that often occurs at an advanced stage of the disease due to the absence of symptoms in the early stages. Early detection of lung cancer is, however, of utter importance to improve survival rates [2]. One of the most promising techniques for early detection is low-dose computed tomography (LDCT) [3]. Studies have already shown that LDCT reduces lung cancer mortality by 20% compared to radiography [4]. The NELSON trial reported an even higher reduction of 24–33%, depending on gender, compared to no screening [5]. Nevertheless, this technique does not come without its pitfalls. One of the main challenges of LDCT is the distinction between benign and malignant lesions, resulting in overdiagnosis and high false-positive rates in individuals without lung cancer [6]. The data obtained in the National Lung Screening Trial (NLST) also revealed some additional concerns: in order to reach the desired benefit, i.e., a successful case of early lung cancer detection in a high-risk population, at least 320 high-risk individuals should be screened for three rounds [7]. To achieve this, many patients have to be exposed to the emotional stress and physical risks associated with screening, such as radiation carcinogenesis. Another important technique for lung cancer detection is positron emission tomography (PET), in which quantitative images of regional in vivo biology are obtained by means of 18F-fluorodeoxyglucose (^18^F-FDG) tracer uptake [8]. This technique is characterized by a high sensitivity of 96%, indicating that almost no lung lesions remain undetected [9]. Nevertheless, an increased ^18^F-FDG uptake is not directly related to cancer. Several benign PET-positive lesions, such as inflammation or tuberculosis, are also characterized by an increased ^18^F-FDG signal. Therefore, misdiagnosis and false-positive results arise as a result of misinterpretation [10]. As a result, about 10% of the patients with non-metastasized PET-positive lung lesions underwent unnecessary surgical interventions [11]. Many efforts have been made to solve these issues, such as changing tracers or measuring protocols, but all alternatives were found to be less appropriate in daily clinical practice [12].

Due to all these challenges, researchers are still looking for early biomarkers that can be used to complement existing screening techniques to provide a safe, faster, and more accurate diagnosis. Biomarkers are defined as “characteristics that are objectively measured and evaluated as indicators of normal biological processes, pathological processes, or pharmacological responses to therapeutic interventions” [13] (p. 344). A new platform that has evolved over recent years for identifying potential early-stage biomarkers is metabolomics, which is defined as the analysis of a large number of metabolites in biological fluids [14]. Since cancer cells are rapidly growing, and therefore have an increased energy consumption, metabolic reprogramming occurs as soon as the disease arises. These metabolic changes eventually result in altered metabolite concentrations in the plasma of cancer patients compared to those found in the plasma of healthy controls [15].

Disruptions are found in several metabolic pathways including the tricarboxylic acid (TCA) cycle, glycolysis, oxidative phosphorylation (OXPHOS), and lipid and amino acid metabolism [16]. One of the most discussed metabolic changes that occurs within cancer cells is the ‘Warburg effect’, which was originally characterized by an increased secretion of lactate even in the presence of normal oxygen concentrations [17]. This phenomenon was attributed to a defective OXPHOS or mitochondrial function, assuming that the mitochondrial metabolism only contributes little to energy production. More recent research has however shown that most cancer cells have functional mitochondria, allowing for the upregulation of both glycolysis and OXPHOS in order to fulfill the high anabolic demands of cancer cells [15]. Another fundamental metabolite that supports cancer cell proliferation is the most abundant amino acid in human blood: glutamine. Glutamine serves as an alternative carbon source for energy and building block production [15]. In the case of mitochondrial dysfunction, glutamine also plays an important role as its oxidation and carboxylation replenish the TCA cycle, allowing for further biosynthetic reactions to take place, such as the formation of fatty acids. Malignant cells are characterized by an upregulation of components of fatty acid synthesis, used for the formation of lipid bilayers and an increased amount of saturated fatty acids in the cell membrane to resist oxidative damage [15]. All of the above-mentioned processes ultimately lead to altered metabolite concentrations, resulting in a unique metabolite profile.

One of the main tools that is currently used for metabolic profiling, besides mass spectrometry, is proton nuclear magnetic resonance (^1^H-NMR) spectroscopy since the ^1^H nucleus is omnipresent in metabolites [16]. ^1^H-NMR has several advantages over mass spectrometry such as high reproducibility (>98%), short measuring time, and simple sample pre-treatment. As the detected peak areas correlate directly with the metabolite concentration, this technique also allows for the accurate quantitative analysis of substances even in complex mixtures such as plasma [16]. In this review, several key breakthroughs in the search for early-stage biomarkers in the plasma of lung cancer patients by ^1^H-NMR will be discussed, and the altered metabolic pathways of these biomarkers will be discussed in the context of lung cancer. Additionally, the potential of these metabolites during screening and diagnosis, as well as their functionality during prediction and follow-up, will be shown.

This review presents the evolution of metabolomics studies in which NMR spectroscopy was used to discover metabolic biomarkers in the plasma of lung cancer patients. This evolution is first described systematically by the studies presented in parts I, II, and III. Then, metabolites found to serve as biomarkers for early-stage NSCLC will be discussed, along with their biochemical pathways in the context of the onset of impaired cancer metabolism.

In part I, the feasibility of lung cancer detection via blood plasma using NMR spectroscopy is demonstrated. First, a study by Louis et al. is discussed, in which lung cancer patients were distinguished from healthy controls based on their metabolite plasma profiles [18]. Based on this study, a second study was performed in which lung cancer plasma profiles were compared with the cancer plasma profiles of breast cancer [19]. These studies demonstrate the feasibility of using NMR spectroscopy to use plasma biomarkers to diagnose lung cancer and to differentiate between different cancer types.

In part II, Derveaux et al. used a higher field strength NMR (600 MHz instead of 400 MHz) and the addition of a human serum albumin-binding (HSA-binding) competitor to further improve resolution and sensitivity and to avoid the binding of specific metabolites to HSA, respectively [20]. In this study, the metabolite plasma profiles of lung cancer patients and healthy controls were again compared, but now with the addition of TSP as an HSA-binding competitor to the sample. This is crucial to prevent the binding of plasma metabolites to HSA, an abundant protein in human plasma, and thus allows for a more accurate determination of the plasma metabolite concentrations. Another important breakthrough shows that, in the presence of TSP, maleic acid (MA) can be used as an internal standard. While peak area normalization is often performed in the total spectral area of the sum of all metabolites, the single signal of MA can now be used. The combination of a stronger magnetic field and an improved sample preparation (i.e., the addition of TSP as an HSA-binding competitor and MA as an internal standard) thereby allows for a more accurate analysis of the altered metabolism in cancer.

Part III highlights a second study by Derveaux et al. that focuses on a deeper metabolic interpretation of early-stage lung cancer by comparing pre- and postoperative plasma metabolite profiles of patients diagnosed with early-staged NSCLC [21]. Importantly, pre- and postoperative plasma samples were donated by the same patients at different timepoints before and after complete surgical resection (i.e., lobectomy). Collection of plasma samples via this longitudinal study design leads to the reduction in patient-to-patient variation and the specification of the plasma metabolic phenotype for early-stage lung cancer. This study shows the potential of plasma biomarkers to (i) monitor therapy/surgery response and (ii) detect early disease recurrence in lung cancer.

A graphical overview of these three parts is presented as Figure A1 in Appendix A. 

## 2. Evolution in NMR Metabolomics Research—Part I: Feasibility of Lung Cancer Detection via Blood Plasma Using NMR Spectroscopy

### 2.1. Feasibility: Lung Cancer Detection

#### 2.1.1. Study Set-Up

The first study by Louis et al. investigated whether lung cancer patients could be distinguished from healthy controls based on the plasma metabolic phenotype with the use of ^1^H-NMR [18]. A total of 357 lung cancer patients and 347 controls, defined as patients with noncancerous disease, were included. If all inclusion criteria were fulfilled, fasting venous blood samples were collected and processed to obtain plasma aliquots for ^1^H-NMR analysis. The obtained NMR spectrum was segmented into 110 spectral regions based on metabolite spiking, in which a small quantity of known metabolites was added to the sample to identify the peak positions [19]. This was essential since metabolomics was a relatively new discipline, with a lack of reference NMR spectra for all known metabolites [22]. This method is preferred to obtain an optimal signal assignment certainty. These 110 normalized integration regions were the variables used for multivariate statistics. The integration regions are the variables for the multivariate statistics, and the value of each variable is determined by the area under the peak of the integration region (and therefore directly correlated with the concentration of a specific metabolite). The NMR measurements were carried out using a 400 MHz NMR spectrometer. More technical details can be found in Appendix B or in the paper of Louis et al. [18].

For the statistical analysis, both lung cancer patients and controls were divided into a training and an independent validation cohort. Subdivision into these two groups is of great importance since external validation is essential to provide reliable conclusions and to confirm, or refute, the strength of the statistical model. Prior to this external validation, a 7-fold internal cross-validation was carried out in the training cohort where validation occurs repeatedly using different subsets of the same dataset. Before subdivision occurred, a principal component analysis (PCA) was carried out to identify confounders. Some outliers were excluded based on Hotelling’s T2 range test, resulting in 331 lung cancer patients and 315 controls. From these cohorts, 233 lung cancer patients and 226 controls were assigned to the training cohort, leaving 98 lung cancer patients and 89 controls for the validation cohort (Table 1). The most common methods used for multivariate statistics include PCA, partial least squares projection to latent structures (PLS), orthogonal projection to latent structures (OPLS), and orthogonal partial least squares discriminant analysis (OPLS-DA). OPLS-DA was used in the studies described here, which is common in NMR-metabolomics. The obtained statistical model can then be interpreted using different model parameters: R2X (the goodness of fit; variation within the groups explained by the model; indicated on the predictive vertical axis), R2Y (explanation between the groups; indicated on the horizontal axis), and Q2 (the predictive accuracy of the trained model) [21]. The closer the values of R2X and R2Y to 1, the better the model. A Q2 value > 0.30 is considered a highly predictive model in the context of metabolomics [21]. Training of the classification model is always performed using data from the training cohort, while additional independent validation of the model is always performed using data from the independent validation cohort, where no group information is provided. High sensitivity and specificity rates in this validation cohort will, therefore, indicate the actual differentiation between the two groups [21]. The variables responsible for the observed differentiation can be analyzed using the variable of importance of projection (VIP) analysis. Another important parameter that is taken into account is the area under the curve (AUC) value of the receiver operating curves (ROCs), which indicates the overall performance of both training and validation models. It is assumed that the closer the AUC value to 1, the better the model. More information about OPLS-DA analysis can be found in Appendix B or in the book of Eriksson et al. [23].

#### 2.1.2. Results

As mentioned above, an OPLS-DA model was first trained using the training cohort (Table 1). This resulted in a model that was able to distinguish the metabolite plasma profiles of the lung cancer patients from those of the controls with a sensitivity of 78% and a specificity of 92% (Figure 1a) (Table 1). Since the trained OPLS-DA model showed an R2X and R2Y of 0.864 and 0.477, respectively, and a Q2 value of 0.391, a good overall quality and predictive accuracy can be assumed (Table 1). The statistical model was then applied to the independent validation cohort, showing a sensitivity and specificity of 71% and 81%, respectively (Figure 1b) (Table 1). The overall performance of the training and validation models was confirmed by AUC values of 0.88 and 0.84, respectively (Figure 1c) (Table 1).

#### 2.1.3. Supporting Evidence

Puchades-Carrasco et al. analyzed whether serum metabolic profiling could be used to diagnose lung cancer [24]. Serum samples were collected from 182 lung cancer patients (both early and advanced stages) and 87 healthy controls. First, PCA was performed to evaluate intrinsic variability within the data, clustering trends, and potential outliers. Afterward, OPLS-DA was carried out using a training cohort of 142 lung cancer patients and 74 controls, and a validation cohort of 40 lung cancer patients and 13 controls. This study showed that serum metabolic profiling allowed for the differentiation between lung cancer patients and healthy controls with a sensitivity and specificity of 92.3% and 95%.

Rocha et al. investigated the plasma metabolite profiles of 85 primary lung cancer patients and 78 healthy controls [25]. PCA analysis was first performed to detect clusters and outliers within the dataset, followed by OPLS-DA. Monte Carlo Cross Validation (MCCV) was used to check the predictive ability of the model by repeatedly using different subsets of the same dataset [26]. Some disadvantages of this method include its computational intensity, the risk of overfitting, and subsampling bias which jeopardizes the model’s validity and reliability [27]. In the study, a sensitivity and specificity of 92% and 89% were reached, together with a Q2 score of 0.64. The results confirm the potential of plasma metabolite profiling for lung cancer diagnosis and screening.

A study by Carrola et al. further supported these findings using urine instead of plasma [28]. Urine samples from 71 lung cancer patients and 54 healthy controls were compared. PCA was applied to detect clusters and outliers, followed by OPLS-DA and MCCV analyses. Samples from lung cancer patients and healthy controls could be distinguished, with a sensitivity and specificity of 93% and 94%, respectively, together with an overall classification rate of 93.5%.

### 2.2. Feasibility: Differentiation between Cancer Types: Lung Cancer versus Breast Cancer

Previous experiments demonstrated the feasibility of plasma metabolite alterations to differentiate lung cancer patients from healthy controls. This raised the question of whether the metabolic fingerprint was a general cancer marker or whether it could also be used to differentiate between different types of cancer. In a second study by Louis et al., it was demonstrated that the plasma metabolite profile allows one to differentiate between lung and breast cancer [19].

#### 2.2.1. Study Set-Up

A total of 145 lung cancer patients with adenocarcinoma and 147 breast cancer patients with adenocarcinoma were enrolled in the study. If all inclusion criteria were fulfilled, fasting venous blood samples were collected and processed in order to obtain plasma aliquots used for ^1^H-NMR analysis. The NMR analysis is as described above in Section 2.1.1, resulting in 110 integration values for the statistical analyses. More technical details can be found in Appendix B or in the paper of Louis et al. [19].

After removing outliers (Hotelling’s T2 test), a total of 135 lung cancer patients and 140 breast cancer patients were subdivided into a training and validation cohort. The training cohort consisted of 54 female lung cancer patients and 80 breast cancer patients, leaving 81 male lung cancer patients and 60 breast cancer patients for the validation cohort (Table 2). Male and female lung cancer patients were allocated in separate groups to exclude a confounding effect of gender.

#### 2.2.2. Results

The OPLS-DA classification model was first trained using the training cohort. This resulted in a sensitivity of 93% (93% of the lung cancer patients were correctly classified) and a specificity of 99% (99% of the breast cancer patients were correctly identified), indicating that almost all female breast cancer patients were correctly identified. This was further confirmed by an AUC of 0.96 (Table 2) (Figure 2a,b). The predictive accuracy was then determined using the independent validation cohort, showing a sensitivity of 89% and specificity of 82%, together with an AUC of 0.94 (Figure 2b,c).

Another study by Louis et al. further confirmed the ability of the metabolic fingerprint to differentiate between different cancer types, where breast, colon, and lung cancer were successfully differentiated [29]. Here, 95% of the breast cancer patients, 78% of the colorectal cancer patients, and 84% of the lung cancer patients were correctly identified.

## 3. Evolution in NMR Metabolomics Research—Part II: Developments in Preanalytical Sample Preparation and NMR Measurement Procedure

The studies discussed above have shown the feasibility of plasma metabolic biomarkers as a tool for screening and diagnosis of lung cancer, as well as for the differentiation between different cancer types. These studies were all performed using a 400 MHz NMR spectrometer. Since these experiments were feasible, i.e., they were able to distinguish the lung cancer patients from controls based on plasma metabolic phenotypes, it was worth investing in a stronger magnet of 600 MHz with higher sensitivity. The higher resolution allows for the definition of more specific integration regions, i.e., more integration regions that contain a signal representing one single metabolite, eventually resulting in a more accurate statistical analysis and identification of the most differentiating metabolites. While distinguishing plasma metabolite profiles of lung cancer and controls was initially the primary focus, the use of a stronger magnetic field could contribute more to a better understanding of impaired cancer metabolism.

### 3.1. TSP as an HSA Binding Competitor and MA as an Internal Standard

Although several metabolomics studies have identified alterations in metabolite concentrations within the plasma of lung cancer patients, an essential factor has often been overlooked. Derveaux et al. demonstrated that, since several plasma metabolites have a high affinity for the plasma protein HSA, only the HSA-unbound fraction of these metabolites is measured [20]. In other words, the ^1^H-NMR signal intensities of these metabolites do not reflect their total concentration, and this will, therefore, lead to an underestimation of the total plasma metabolite concentration [20]. In addition, this study showed that some metabolites are not only characterized by this affinity towards HSA but that the commonly used internal (chemical shift) reference TSP also strongly binds to HSA. Therefore, TSP cannot be used as an internal reference since the obtained signal no longer reflects the amount added during the sample preparation [30].

However, TSP has another major advantage that has not been described before. Since TSP has a much higher affinity for HSA than the other plasma metabolites, its addition leads to the saturation of the HSA metabolite-binding sites and thus to the dissociation of these metabolites from HSA (Figure 3). TSP could, therefore, be used as an HSA-binding competitor: by preventing the binding of other metabolites to HSA, the total metabolite plasma concentrations can be measured. It can, therefore, be concluded that TSP addition is crucial for the accurate determination of total plasma metabolite concentrations in metabolomics studies.

Additionally, due to its high affinity for HSA, TSP is not a suitable candidate as the internal standard for quantification, indicating that another internal standard should be used in order to quantify plasma metabolites. Derveaux et al. proposed MA as a suitable candidate for absolute quantification and peak area normalization since it gives rise to a sharp, isolated signal (i.e., around 6 ppm in the ^1^H NMR spectrum) without overlapping with other metabolite signals [20]. In addition, MA can be purchased with excellent purity, it is easily dried, and it has a distinct solubility in D_2_O [20]. To make sure that MA could be used as an internal standard, the affinity towards HSA was analyzed. Here, it was shown that MA showed some affinity towards HSA but, as for several other metabolites, at a lower degree compared to TSP. Therefore, when added together with TSP, MA did not bind to HSA, enabling it to be used as a reliable internal standard for quantification. Knowing this, the plasma metabolite profiles of lung cancer patients were compared with those of controls in the presence of 4 mM TSP as an HSA-binding competitor and MA as the internal standard.

### 3.2. Study Design

In order to validate the new proposed methodology, a total of 141 lung cancer patients and 135 healthy controls were included. If all inclusion criteria were fulfilled, fasting venous blood samples were collected as described before and processed to obtain plasma aliquots for ^1^H-NMR analysis. This led to a total of 114 lung cancer patients and 118 healthy controls for further analysis. In contrast to previous studies, the samples were enriched with the addition of 4 mM TSP as the HSA-binding competitor and 53.58 µM MA as the internal standard. Similarly to the previous studies, metabolite spiking was performed to divide the NMR spectrum into well-defined integration regions. Metabolite spiking with 62 metabolites resulted in the definition of 237 integration regions in the ^1^H-NMR spectrum recorded on a 600 MHz spectrometer [20]. These 237 normalized integration regions were the variables used for multivariate statistics. More technical details can be found in Appendix B or in the paper of Derveaux et al. [20].

Both patient groups were subdivided into a training and validation cohort, where the training cohort consisted of 80 lung cancer patients and 80 controls, and the validation cohort consisted of 34 lung cancer patients and 38 controls (Table 3).

### 3.3. Results

Data reduction was performed (cross-validation; ‘jack-knifing’), leaving 78 variables for final statistical analysis. Using these 78 variables, an unsupervised PCA plot was created to gain some first insights into the separation of the data. A clear distinctive trend was observed between the plasma metabolite profiles of the lung cancer patients and controls within the training cohort (Figure 4a). Afterward, the same 78 variables were used to train a supervised OPLS-DA model. The constructed model was able to distinguish the two groups with a sensitivity of 85% and a specificity of 93% (Table 3) (Figure 4b). The strength of this model was further confirmed by high values of the R2X (Cum), R2Y (Cum), and Q2 (Cum) parameters of 0.861, 0.581, and 0.36, respectively (Table 3). The overall performance was also assessed by the ROC curve, where an AUC value of 0.95 was obtained (Table 3) (Figure 4c). Afterward, the obtained results were validated via the independent validation cohort. The ability to distinguish the two groups remained, with a sensitivity and specificity of 74% each (Table 3) (Figure 4d). The proposed methodology allowed for (i) the measurement of actual metabolite concentrations using TSP and (ii) the quantification of absolute plasma metabolite and normalization of peak area using MA. This enabled a precise study of the plasma metabolite profile in order to identify potential biomarkers.

Therefore, further analysis was performed to identify the most significant contributors to the observed discrimination between the two groups. The most significant decreases in the plasma of lung cancer patients included lipid signals from fatty acid chains, phosphatidylcholines, and sphingomyelins. The most significant increases in the plasma of lung cancer patients included glucose, isoleucine, glycerol, and isopropanol [20]. The decrease in the above-mentioned lipids has been widely discussed in literature and is attributed to various factors including enhanced membrane synthesis, increased enzyme activity, and precursor consumption in lung cancer.

## 4. Evolution in NMR Metabolomics Research—Part III: Plasma Biomarkers for Early-Stage NSCLC and Their Potential for Detection and Monitoring of NSCLC Recurrence

Derveaux et al. improved metabolite quantification and the metabolite fingerprinting of lung cancer by adding a more accurate internal standard (MA) in combination with an HSA-binding competitor TSP and using a stronger NMR magnetic field strength [20]. However, this can even be further improved by reducing patient-to-patient (donor-intrinsic) variabilities and modeling only the metabolic changes that are related to the onset of lung cancer. To overrule these variabilities, Derveaux et al. performed another study with a longitudinal setup in which the pre- and postoperative plasma metabolite profiles of the same individuals, all early-stage NSCLC patients who received surgery, were compared [21]. Analysis of the pre- and postoperative plasma samples provides early-stage NSCLC and healthy metabolite profiles, respectively. As each patient can now serve as its own control, paired multivariate statistics can be used to differentiate pre-and postoperative plasma metabolite profiles where patient-to-patient variations are reduced. The most differentiating plasma metabolites in these models are, therefore, very specific biomarkers for early-stage NSCLC. This unique metabolomics study shows the potential of early detection of NSCLC recurrence by non-invasively monitoring NSCLC patients during therapy follow-up via blood samples.

### 4.1. Study Design

A total of 114 lung cancer patients diagnosed with resectable stage I-IIIA non-small cell lung cancer (NSCLC) were included. In case all inclusion criteria were fulfilled, fasting venous blood samples were collected at three different time points, including baseline (B; day of surgery; preoperative), control (C; after diagnosis but before surgery; preoperative), and effect (E; three months after surgery; postoperative-3M). For the effect group, it was clinically confirmed that none of these patients showed disease recurrence after three months. All blood samples were processed to obtain plasma aliquots for ^1^H-NMR analysis. The same NMR procedure was applied as proposed in the previous study of Derveaux et al., where (i) a 600 MHz NMR spectrometer was used, (ii) TSP was added as an HSA-binding competitor, and (iii) MA was added as a quantitative internal standard [20]. Again, the NMR spectrum was divided into 237 well-defined integration regions which were used as variables for further statistical analyses. In case (i) no complete surgical resection could be performed, (ii) a postoperative pathological non-malignant diagnosis appeared, or (iii) disease recurrence occurred within a period of 6 months, patients were excluded from the study. This resulted in a total of 74 remaining early-stage lung cancer patients (Table 4) [21]. More technical details can be found in Appendix B or in the paper by Derveaux et al. [21].

All patients were subdivided into a training cohort (50 patients) and an independent validation cohort (24 patients) (Table 4). Since three different comparisons were aimed for (B vs. E; C vs. E; B vs. C), three different statistical models were constructed. The third comparison, B vs. C, was carried out as an additional control to find out whether the metabolite profiles at these two preoperative time points are similar and patient-specific.

### 4.2. Results

A first supervised OPLS-DA model was trained with the data from the baseline (B) and effect (E) timepoints of the training cohort. Here, excellent discrimination between pre- and postoperative profiles was observed with a sensitivity and specificity of 92% (92% of the samples were correctly identified as preoperative) and 96% (96% of the samples were correctly identified as postoperative), respectively (Table 4) (Figure 5a). The predictive accuracy was further strengthened by the high values of the obtained R2X (Cum), R2Y (Cum), and Q2 (Cum) parameters of 0.55, 0.67, and 0.42, respectively (Table 4). These results were then validated in the independent validation cohort, resulting again in an excellent distinction of pre- and postoperative plasma metabolite profiles with a sensitivity and specificity of 88% and 92%, respectively (Table 4) (Figure 5b). Additionally, excellent AUC values of 0.99 and 0.97 were reached within the training and validation cohort, respectively (Table 4) (Figure 6). While OPLS-DA is a very strong method for modeling differentiation between the two groups, this method is not appropriate for dependent, paired samples. Since this study compared plasma samples derived from the same lung cancer patient at different timepoints, OPLS-effect projection (OPLS-EP) studies, which take the paired character of the samples into account, could also be performed. Therefore, an OPLS-EP model was trained and validated as well. Here, even better parameter values of the R2X (Cum), R2Y (Cum), and Q2 (Cum) were found, reaching values of 0.59, 0.89, and 0.76, respectively (Table 4). These values strongly support the differentiation found between pre- and postoperative metabolite profiles. More information about OPLS-EP analysis can be found in Appendix B and in the paper by Jonsson et al. [31].

After comparing B/E timepoints, the second OPLS-DA and OPLS-EP models were trained with the data derived from the control (C) and effect (E) timepoints of the training cohort. Again, the constructed model succeeded in distinguishing pre- and postoperative plasma metabolite profiles with outstanding parameters for both OPLS-DA and OPLS-EP. OPLS-DA showed a sensitivity and specificity of 88% and 90%, respectively, together with R2X (Cum), R2Y (Cum), and Q2 (Cum) parameters of 0.53, 0.61, and 0.36, respectively (Table 4) (Figure 5c). Validation of the OPLS-DA model resulted in even higher sensitivity and specificity rates of 96% and 91%, respectively (Table 4) (Figure 5d). The excellent performance of the constructed model was confirmed by the high AUC values in both training and validation cohorts of 0.97 each (Table 4) (Figure 6). The OPLS-EP model showed R2X (Cum), R2Y (Cum), and Q2 (Cum) parameters of 0.57, 0.83, and 0.60 (Table 4).

As an additional control, a third OPLS-DA model was constructed using data derived from the baseline (B) and control (C). First, the OPLS-DA model was trained using data from the training cohort. This resulted in low R2X (Cum) and R2Y (Cum) values of 0.31 and 0.15, respectively, and a very low Q2 (Cum) of 0.08 (Table 4). 

In the next step, key contributors to the metabolic shift between the pre- and postoperative profiles were identified. For both the OPLS-DA and OPLS-EP models, the same variables were identified as significant contributors. Preoperative plasma profiles (i.e., lung cancer profiles) showed a decrease in lactate, cysteine, and asparagine levels and an increase in acetate levels. All these changes in concentration were found to be significant (paired *t*-test; *p* < 0.001) (Figure 7) [21].

## 5. Discussion of Breakthroughs in NMR Metabolomics Research

It is noteworthy to mention that the models discussed in parts II and III revealed different sets of discriminating key metabolites with increased/decreased plasma levels compared to other metabolomics studies in literature. For example, lactate plasma levels were found to be decreased in the metabolite profile of lung cancer by Derveaux et al., while other studies report increased plasma lactate in lung cancer patients [25,32]. These different outcomes throughout the literature can be explained because of (i) sample preparation with or without the addition of TSP as an HSA-binding competitor, (ii) inclusion of patients diagnosed with different stages of lung cancer, and (iii) patient-to-patient metabolic variability between groups.

The studies discussed in parts II and III involve the addition of TSP to the sample as an HSA-binding competitor. This pre-analytical optimization is of great importance, as some metabolites possess a strong affinity for HSA (e.g., lactate). As a result, without the addition of an HSA-binding competitor, only the unbound fraction of these metabolites is measured, leading to an underestimation of their total plasma concentrations. Moreover, it is known that the amount of HSA can vary considerably among individuals and time, leading to HSA-dependent differences in measured plasma metabolite levels across different persons and different timepoints.

For example, lactate concentrations can be heavily underestimated without the addition of an HSA-binding competitor and may even lead to the incorrect interpretation of an apparent lower plasma lactate level in healthy controls as compared to lung cancer patients, who typically have a lower HSA concentration. To address this issue, TSP was introduced as a strong HSA-binding competitor. Since TSP exhibits a higher affinity for HSA, its addition results in a saturation of the metabolite-binding sites with TSP and the subsequent dissociation of metabolites from HSA, leading to the correct measurement of the total plasma metabolite concentrations. This is also nicely shown in the study presented in part III, in which lactate was found to be decreased in the preoperative metabolite profile (i.e., the metabolite profile of lung cancer), thereby indicating that the measured lactate concentration without the addition of an HSA-binding competitor rather reflects the differences in HSA concentration between the pre- and postoperative groups and that caution should be taken when describing altered metabolic pathways if no HSA-binding competitor was added to the plasma samples.

A second explanation for interstudy variabilities is related to the normalization method. Many studies performed normalization by calculating the total integration area of the full ^1^H-NMR spectrum and normalizing the integration value of each integration region towards the sample-specific total [20]. However, statistical analysis may be more strongly influenced by concentration differences in highly abundant plasma metabolites as these dominate the total area. Using this normalization method, plasma metabolite concentrations can only be determined relatively. For this reason, the use of an internal standard is recommended for absolute quantification. Derveaux et al. added a known concentration of MA to each sample, as this would allow for normalization towards MA to obtain peak areas independent of the changes in other signals.

A third explanation for the observed variability in the identified key contributing metabolites could be attributed to the differences in the inclusion criteria of the various studies. While the study discussed in part II includes patients with any stage of lung cancer, where a large part was characterized by metastasis (e.g., >25% stage IV patients in the studies presented in parts I and II), the study discussed in part III only focused on early-stage lung cancers. Since advanced disease stages show more metabolic changes, the plasma metabolite concentrations will also differ from those in the early stages. A study by Larkin et al. confirmed this by demonstrating the ability of plasma metabolites to discriminate early stages from metastasized cancer stages [33]. Here, it was shown that patients with metastatic cancer had increased levels of metabolites such as threonine. Since threonine was not considered to be significant in the last study by Derveaux et al., where only early stages of lung cancer were included, observed differences are likely to be caused by advanced lung cancer stages [21]. This also explains why decreased lipid concentrations (sphingomyelin, phosphatidylcholines, and phospholipids) and increased glucose concentrations in lung cancer plasma were found in multiple publications, including the study discussed in part II, which was absent in part III [18,19,25]. These examples, therefore, further highlight the importance of careful interpretation of the changes in the biochemical pathways, which can heavily depend on the cancer stages of the included patients.

A fourth explanation for different findings can be found in patient-to-patient variations. In most metabolomics studies, a group of (lung) cancer patients is compared with a group of healthy controls or with patients diagnosed with a different type of cancer. While this type of study-set up provides a strong confirmation of the feasibility of differentiating two groups, which is needed for applying the metabolite profile in a diagnostic setting, individual follow-up of patients after treatment remains very challenging. For this reason, the study of Derveaux et al., described in part III, compares pre- and postoperative plasma metabolite profiles of early-stage NSCLC patients via a longitudinal study design (of the same patients). This way, each patient can serve as their own control, thereby greatly reducing patient-to-patient variabilities. The effect of this longitudinal sample collection was first confirmed by comparing the pre-and postoperative metabolite profiles (i.e., the metabolite signatures of lung cancer and healthy condition, respectively) via OPLS-DA analysis, where the paired character of the samples was not taken into consideration yet. The success of a further reduction in patient-to-patient variation was confirmed using OPLS-EP, in which the paired character of the samples was taken into account.

The importance of all the methodology improvements discussed above is, for example, demonstrated when analyzing the lactate plasma levels in lung cancer. While the lactate concentration in the plasma of lung cancer patients is shown to be decreased in the most recent study by Derveaux et al., contradictory results are found throughout the literature. Zhang et al. compared the metabolite profile of serum from lung cancer patients with healthy controls using a non-targeted ^1^H-NMR approach and a targeted rapid resolution liquid chromatography [32]. Here, lactate was identified as a potential diagnostic biomarker, but with an increase in concentration in lung cancer. Rocha et al. also found an increased lactate concentration in the plasma of lung cancer patients [25]. The first explanation for these contradictory results is the absence of an HSA-binding competitor. Derveaux et al. demonstrated the high affinity of lactate for HSA, resulting in the strong binding of lactate to HSA [20]. Since lung cancer patients are known to have decreased HSA concentrations, less lactate is able to bind, resulting in an apparent higher concentration in the plasma or serum of lung cancer patients compared to healthy controls with more HSA, and therefore less unbound lactate [34]. The results of these studies, therefore, reflect the decrease in plasma HSA concentration of lung cancer patients, rather than the actual total plasma lactate concentration. An additional reason for these contradictory results can be found in the different disease stages of the included lung cancer patients. While the most recent study by Derveaux et al. only included early-stage NSCLC, some studies included all stages of lung cancer [21]. Puchades-Carrasco et al. compared metabolic biomarkers between early-stage and advanced-stage NSCLC, in which lactate was found to be significantly higher in advanced stages compared to early disease stages, showing the importance of the disease stage in the metabolite profile [24]. Besides disease stage, disease type should also be considered. Zhang et al. included lung cancer patients with either adenosquamous-, adeno-, or small-cell carcinomas [32]. Kowalczyk et al. studied whether the tissue metabolite profile could be used to discriminate between different types of lung cancer and were able to distinguish between adeno- and small-cell carcinoma [35]. A distinction could also be made using blood plasma, although other metabolites seemed to contribute to the observed differences in metabolite profiles. Even though lactate was not identified as a significant contributor in tissue, it is unknown whether plasma analysis would identify lactate as an important metabolite in this study [35]. Therefore, caution should be taken regarding the inclusion of patients with different types of lung cancer if the goal is to unravel the adapted metabolic pathways for a certain type, rather than cancer diagnosis.

Altogether, the study discussed in part III combined the three aforementioned advantages: (i) the addition of TSP to the samples as an HSA-binding competitor, (ii) the inclusion of only early-stage NSCLC patients, and (iii) the reduction in patient-to-patient variability by comparing paired pre- with postoperative plasma metabolite profiles. The study clearly revealed four key indicators for the onset of early-stage NSCLC: lactate, asparagine, acetate, and cysteine.

## 6. Metabolic Pathways Involved in (Lung) Cancer

As optimized preanalytical conditions were used in the study of part III, resulting in accurate measurements of total plasma metabolite levels, this section focuses on the changes in biochemical pathways due to in-/decreased plasma metabolite levels that were revealed as key contributors in the changed lung cancer metabolite profile.

### 6.1. Lactate

Cancer cells are characterized by high energy demands due to high proliferation rates, and one of their hallmarks is their ability to reprogram the metabolism, including the glucose and lactate metabolism [36]. Indeed, the study described in part III clearly reveals that lactate levels are decreased in lung cancer plasma. This important finding can be explained by a detailed description of the lung cancer’s metabolism. In typical glycolytic cancer cells, this high energy demand is met by increased glucose uptake via the upregulated levels of glucose transporters (GLUTs), especially GLUT3 and GLUT5 in lung cancer cells (Figure 8) [37]. Unlike in normal cells, where glucose is broken down to two molecules of pyruvate which can enter the TCA cycle to produce at least 32 ATP molecules, cancer cells metabolize glucose in a different way. Cancer cells partially metabolize glucose via the nonoxidative catabolism, even in the presence of sufficient amounts of oxygen. Pyruvate is converted into lactate, preventing its use as an intermediate to form acetyl-CoA to drive the TCA cycle, thereby resulting in the production of only 2 ATP molecules per glucose molecule [38]. This phenomenon is also known as the ‘Warburg effect’ or aerobic fermentation. Even though it was believed that glycolytic cancer cells only used aerobic fermentation, it is known by now that ATP is also produced via oxidative phosphorylation (OXPHOS). Since OXPHOS yields higher levels of ATP production compared to aerobic fermentation (aerobic glycolysis), it is believed that the Warburg effect occurs to support biosynthetic requirements of uncontrolled proliferation, while the OXPHOS produces sufficient energy to support cell growth. This hypothesis is further supported by the fact that a different form of pyruvate kinase (PK) is expressed in lung cancer cells [38]. PK catalyzes the conversion of phosphoenolpyruvate (PEP) into pyruvate, thereby releasing one ATP molecule. In normal cells, the tetrameric form of isoenzyme PKM2 is expressed, which is characterized by a high affinity for PEP. Lung cancer cells, however, express the dimeric form of PKM2, which has a much lower affinity for PEP (Figure 8). Therefore, all glycolytic intermediates preceding PKM2 activity will accumulate and be used for biosynthetic processes to stimulate tumor proliferation such as nucleotide synthesis [38] (Figure 8). This does not imply that pyruvate will no longer be formed at all, but rather to a lesser extent. Pyruvate will then either be converted into lactate or enter the OXPHOS as discussed earlier. The upregulated conversion to lactate in glycolytic cancer cells further leads to increased amounts of intracellular lactate production and excretion via monocarboxylate transporter 4 (MCT4) (Figure 8). This increased excretion of lactate into the microenvironment results in acidification, immune cell function modulation, extracellular drug deactivation, and promotion of invasion and metastasis [36,38]. Lactate, produced by the glycolytic cancer cells, can, however, be taken up from the microenvironment by oxidative cancer cells via MCT1 transporters in order to be used as a substrate for increased gluconeogenesis since it is the primary source of a gluconeogenic precursor [39,40]. This transport of lactate between glycolytic and oxidative cancer cells is also referred to as the “lactate shuttle” [41]. Studies have shown that one of the key enzymes involved in gluconeogenesis, phosphoenolpyruvate carboxykinase (PEPCK), is responsible for the replenishment of glycolytic pools in cancer cells, thereby supporting cancer cell growth. Leithner et al. showed that one of the isoforms of this enzyme, PCK2 (or PEPCK2), was shown to be actively expressed by lung cancer cells and that expression levels were enhanced under low-glucose conditions [39]. This principle was confirmed since PEPCK inhibition led to a reduction in tumor growth [42]. The gluconeogenesis is even further supported in lung cancer by the upregulation of pyruvate carboxylase (PC) since this enzyme is responsible for the transformation of pyruvate to oxaloacetate [43]. Oxaloacetate will eventually be converted to PEP by the increased levels of PCK2. In this way, oxidative lung cancer cells can convert lactate into PEP via the upregulation of PC and PCK2 (Figure 8). PEP can eventually donate its carbons to the gluconeogenesis pathway, thereby providing energy in a glucose-independent manner. This process, therefore, reflects the metabolic symbiosis between glycolytic and oxidative tumor cells in order to survive and stimulate cancer growth.

Besides the fact that lactate can be used as an energy source and precursor for gluconeogenesis, it can also be used as a signaling molecule for tumor survival and progression. Lactate is able to bind to the membrane-bound receptor G protein-coupled receptor 81 (GPR81), which is typically upregulated in cancer cells [44] (Figure 8). This upregulation in lung cancer cells is caused by lactate via the signal transducer and activator of transcription 3 (STAT3). Lactate upregulates the transcriptional factor Snail, resulting in the assembly of the Snail/EZH2/STAT3 complex, which enhances STAT3 activity. STAT3 will in turn bind to the GPR81 promotor, activating its expression in cancer cells [45]. The binding of lactate to this receptor could eventually lead to the activation of the phosphatidylinositol 3-kinase (Pi3K)/protein kinase B (Akt) pathway, resulting in the release of amphiregulin, which stimulates angiogenesis of tumor endothelial cells, tumor cell growth, and proliferation (Figure 8) [44]. Another consequence of GPR81 binding in lung cancer is the induction of the programmed cell death protein 1 ligand (PD-L1), causing tumor cell immune escape [46] (Figure 8). Therefore, despite increased production and excretion rates, a decrease in plasma lactate concentrations can be observed.

### 6.2. Acetate and Amino Acids Cysteine and Asparagine

The most recent paper by Derveaux et al. showed that lactate, cysteine, acetate, and asparagine were the most significant metabolites in the differentiation between early-stage lung cancer and healthy conditions [21]. Acetate was found to be increased in the plasma of lung cancer patients, while cysteine and asparagine were shown to be decreased. This will be discussed more in detail since the study took the following into account: (i) limited patient-to-patient variability (paired samples); (ii) use of a new methodology that more accurately represents the total metabolite concentrations via the addition of TSP as a binding competitor and MA as an internal standard; and (iii) only early stages of NSCLC (I-IIIa) were included (instead of all NSCLC stages), thereby limiting variability.

Acetate metabolism is an important pathway in many cancers, including lung cancer, since they are characterized by an increased expression of acetate-metabolizing enzymes like acetyl-CoA synthetases [47]. Lung cancer is characterized by an increased expression of acetyl-CoA synthetase 2 (ACSS2), which catalyzes the ATP-dependent ligation of acetate and CoA to produce acetyl-CoA. Acetyl-CoA will then be used for energy production, but also as a substrate for other biochemical reactions such as the synthesis of ketones and sterols [47]. This production of energy substrates enables further stimulation of tumor growth. This was further confirmed by Comerford et al., who showed that ACSS2 knockdown in human patient tumor cell lines led to growth inhibition [48]. Another important enzyme in lung cancer is ATP-citrate lyase (ACLY), a key enzyme of de novo fatty acid synthesis which is responsible for the transformation of TCA-derived citrate into acetyl-CoA [49]. Migita et al. showed an increase in ACLY expression in human lung adenocarcinoma cells, indicating an increased formation of acetyl-CoA, which can be used as a precursor for fatty acid synthesis, cholesterol synthesis, and energy production. The importance of this enzyme was demonstrated by inhibition via RNA interference, which induced tumoral growth arrest. Even though an increased acetate consumption is seen in cancer, an increase in plasma concentration can be observed in lung cancer patients. Research has demonstrated a pathway involved in de novo acetate production from pyruvate, which was pronounced at moments of overflow metabolism [50]. Overflow metabolism is defined as a situation in which the metabolic supply outpaces the metabolic demand, resulting in an accumulation of intermediate products [50]. This is typically observed in cancers, where an incomplete metabolism occurs together with increased glucose uptake by cancer cells, yielding metabolic intermediates, such as acetate, into the extracellular space. This excessive amount of acetate that is released by nutrient-rich tumor cells can eventually be consumed by nutrient-poor tumor cells, allowing them to function and grow in poor vascular environmental conditions. Acetate, therefore, plays a key molecule in the symbiotic interaction that occurs between nutrient-poor and -rich tumor cells [50]. Another explanation for the increased acetate plasma concentration can be found within systemic metabolism. The body itself can also act on cancer development, thereby contributing to altered plasma metabolite concentrations. During cancer, increased levels of acyl-CoA thioesterase 12 (ACOT12), an enzyme which generates acetate from acetyl-CoA, can be found in the liver. This process is activated in conditions of metabolic stress, resulting in the formation and excretion of acetate in the extracellular space [51].

Also, the cysteine plasma concentration shows a significant alteration in lung cancer compared to normal plasma. Contrary to the increased acetate levels, decreased cysteine plasma levels are found in lung cancer. Cysteine is of great importance in energy production, cellular biosynthesis, enzyme catalysis, and redox metabolism [52]. During tumor development, there is an increased demand for cysteine to elevate glutathione production to protect cancer cells from death due to increased, accumulated oxidative stress levels. Cysteine is of great importance, together with glycine and glutamate, for glutathione formation by glutamyl-cysteine ligase (GCL) [52]. Cysteine is transported into the cell using both the excitatory amino acid transporter 3 (EAAT3/SCL1A1) and the alanine–serine–cysteine transporter 1 and 2 (ASCT1/2) [53]. In lung cancer, increased cysteine uptake is enabled by the overexpression of ASCT2 and EAAT3 [54]. Besides its role in ROS protection, cysteine is also used as a carbon source during energy production by contributing to the TCA cycle after its conversion to pyruvate or α-ketoglutarate [55]. Even though many cancer types show an increase in de novo synthesis of cysteine, lung cancer cells show an inactivated pathway during tumorigenesis [52]. Despite the inactivation of cysteine synthesis, cysteine uptake and metabolism remain. Taken together, a decrease in cysteine plasma concentration is observed in lung cancer due to the increased metabolic needs combined with decreased de novo synthesis.

Finally, asparagine was shown to be decreased in the plasma of lung cancer patients. The asparagine metabolism is closely related to the glutamine metabolism. Glutamine is one of the major precursors for de novo biosynthesis of nucleotides as well as other nonessential amino acids [56]. Once glutamine donates its amino group for nucleotide synthesis, in turns into glutamate, which eventually can be used for the synthesis of nonessential amino acid derivatives such as α-ketoglutarate. In this way, α-ketoglutarate can enter the TCA cycle, yielding additional energy for further cancer cell proliferation and survival. Within glutamine-deprived cells, which are frequently observed in cancer cells under physiological conditions, asparagine plays an important role in maintaining cancer cell viability. Just like glutamine, asparagine functions as a precursor for the TCA cycle. Asparagine can be converted into aspartate by asparaginase, which can be converted by aminotransferases into oxaloacetate, which can enter the TCA cycle, providing new energy for cancer cell proliferation. Therefore, asparagine was originally thought to prevent glutamine depletion-induced apoptosis by acting as a precursor substrate for anaplerosis (replenishment of TCA cycle intermediates) [56]. Nevertheless, this hypothesis was challenged as it was shown that inhibition of citrate synthase (CS), the first enzyme of the TCA cycle, prevented glutamine depletion-induced apoptosis. Additionally, asparagine addition did not result in anaplerosis, indicating that the survival-stimulating effect of asparagine was not caused by acting as a precursor for the TCA cycle in cancer cells. Asparagine is, therefore, not catabolized to act as a carbon donor, but it is taken up directly from the extracellular environment to protect cancer cells from glutamine depletion-induced apoptosis and to maintain protein synthesis [57]. Zhang et al. showed that glutamine-deprived cancer cells first undergo cell-cycle arrest, but that translation of the stress-response RNAs, including C/EBP-homologous protein (CHOP), results in the actual activation of cellular apoptosis [56]. To prevent this glutamine depletion-induced apoptosis, asparagine steps in as it is able to suppress the activation of CHOP. The critical role of asparagine was confirmed through the inhibition of asparagine synthetase (ASNS), which resulted in the induction of apoptosis. In addition, Jiang et al. showed that exogenous asparagine is able to induce glutamine synthetase (GLUL) accumulation at a post-transcriptional level, enabling de novo glutamine synthesis [58].

## 7. Conclusions and Future Perspectives

Early-stage plasma biomarkers have great potential to enable early detection of lung cancer, thereby providing a faster diagnosis and improved survival rates. The above-mentioned studies confirmed changes in the plasma metabolite profile upon lung cancer development, enabling to distinguish these patients from the controls. Furthermore, the differentiating OPLS-DA and OPLS-EP models in these studies identified several metabolites as key contributors (i.e., lactate, cysteine, asparagine, and acetate) and therefore as potential plasma biomarkers for lung cancer.

^1^H-NMR is currently used as the golden standard for metabolomics research since it is characterized by multiple advantages such as its simple sample preparation without the need for extractions, noninvasive character, high reproducibility (>98%), short run time, automatization possibilities and only minor matrix effects. The main challenges that remtain include the need for (international) standardized protocols to decrease variabilities caused by alterations in working procedures and to enable a more detailed comparison of study results. An essential step, therefore, is to standardize the incorporation of (i) TSP as an HSA-binding competitor and (ii) a suitable internal standard such as MA. As TSP enables the dissociation of HSA-bound metabolites, representing the actual plasma metabolite concentrations, and an internal standard enables normalization independent from changes in metabolite levels, both practices are required for accurate and quantitative metabolic analysis. Besides the need for standardized protocols, it is crucial to sufficiently use large sample sizes to prevent overfitting when using multivariate statistics, given the large number of variables in metabolomics studies.

These metabolomic studies led to several advantages, i.e., the ability to diagnose lung cancer in a non-invasive, efficient manner, and potential future applications of the plasma metabolite changes were shown throughout the different studies discussed. Louis et al. were able to show the potential of plasma metabolites as biomarkers for lung cancer diagnosis and screening since clear differences were observed between plasma metabolite profiles of lung cancer patients compared to controls [18]. Additionally, the discriminating power of plasma metabolites was shown by their ability to distinguish lung cancer from breast cancer based on different metabolic alterations [19,29]. The last study by Derveaux et al. showed a clear “metabolic shift” in plasma metabolite concentrations between pre- and postoperative states [21]. In general, we aim to use ^1^H-NMR to provide support in three stages of clinical decision making: (i) screening via the combination of ^1^H-NMR and LDCT, (ii) ^1^H-NMR to complement PET, and (iii) the use of ^1^H-NMR for disease prediction and monitoring. Regarding the first aim, we would like to combine ^1^H-NMR and LDCT for screening a large high-risk population. Secondly, we would like to use ^1^H-NMR to complement the PET scan. As PET scans often fail to differentiate between malignant and non-malignant PET-positive lesions, a high number of false positives is observed. Combining PET scans with ^1^H-NMR could overcome this by taking the plasma metabolomics into account. Thirdly, we would ultimately also like to use ^1^H-NMR as a tool for disease prediction and monitoring. Since the plasma metabolite profile changes after disease recovery, plasma biomarkers are a promising, complementary tool to evaluate therapy response, follow-up, and disease recurrence. Plasma biomarkers could even be used in the future to identify disruptions in metabolic pathways with greater detail, allowing for targeted therapeutic interventions as knowledge of altered metabolic processes continues to expand. In summary, it becomes increasingly clear that NMR metabolomics might be able to pave the way towards providing support in clinical decision making regarding early cancer diagnosis via screening (of encumbered populations), complementing PET, assessing therapy response and follow-up monitoring, and potentially predicting disease recurrence.

## Figures and Tables

**Figure 1 ijms-25-04690-f001:**
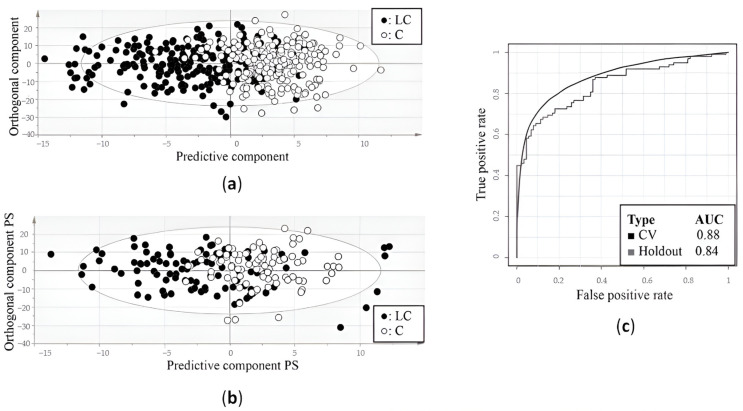
Differentiation between lung cancer patients and healthy controls based on metabolic profiling in plasma. (**a**) Orthogonal partial least squares discriminant analysis (OPLS-DA) score plot from the training cohort shows a distinction between the plasma metabolite profiles of lung cancer patients and controls with a sensitivity of 78% and a specificity of 92%. (**b**) OPLS-DA score plot from the independent validation cohort shows a distinction between the metabolite profiles of lung cancer patients and controls with a sensitivity of 71% and specificity of 81%. (**c**) Receiver operating curves show the high predictive accuracy of the OPLS-DA model of the training cohort for both cross-validation and independent validation, reaching AUC values of 0.88 and 0.84, respectively. AUC, area under the curve; C, controls; CV, cross-validation; LC, lung cancer patients; PS, predicted scores. Figure adapted from reference [18].

**Figure 2 ijms-25-04690-f002:**
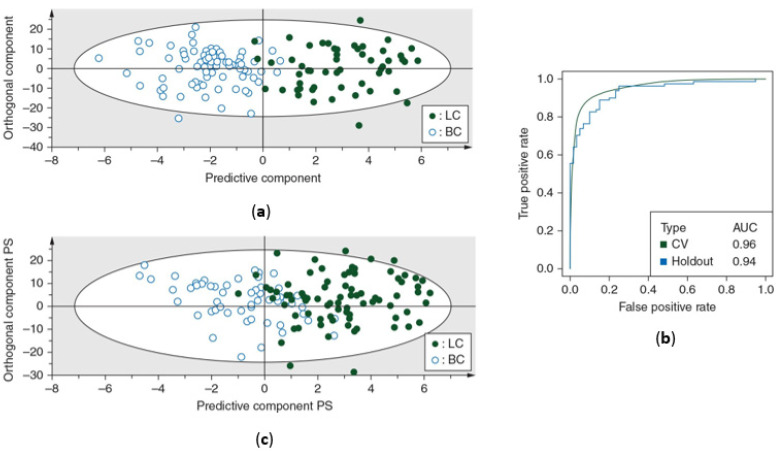
Differentiation between lung and breast cancer patients based on metabolic profiling in plasma. (**a**) Orthogonal partial least squares discriminant analysis (OPLS-DA) score plot from the training cohort shows a distinction between the metabolite profiles of lung cancer patients and breast cancer patients with a sensitivity of 93% (93% of the lung cancer patients were correctly classified) and a specificity of 99% (99% of the breast cancer patients were correctly classified). (**b**) Receiver operating curves show the high predictive accuracy of the OPLS-DA model of the training cohort for both cross-validation and independent validation, reaching AUC values of 0.96 and 0.94, respectively. (**c**) OPLS-DA score plot from the independent validation cohort shows a distinction between the metabolite profiles of lung cancer patients and breast cancer patients with a sensitivity of 89% and specificity of 82%. AUC, area under the curve; BC, breast cancer patients; CV, cross-validation; LC, lung cancer patients; PS, predicted scores. Figure adapted from reference [19].

**Figure 3 ijms-25-04690-f003:**
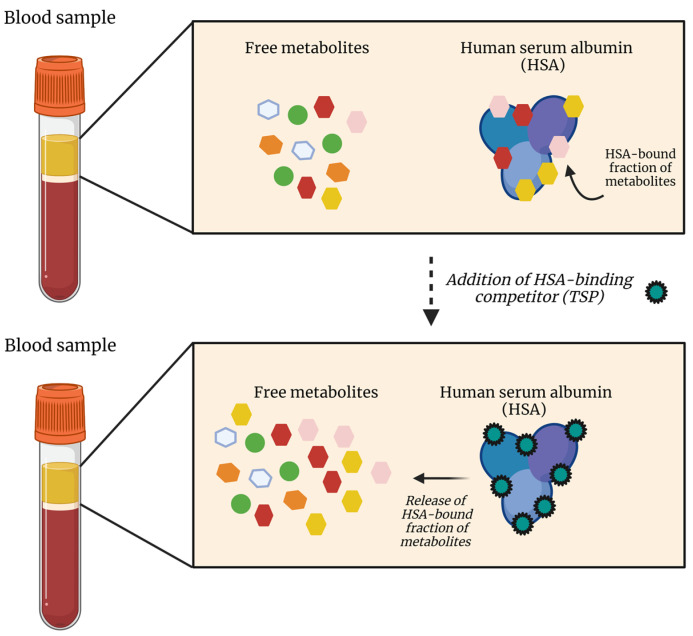
The addition of TSP leads to the dissociation of the HSA-bound fraction of metabolites due to the extremely high affinity of TSP for HSA. While some plasma metabolites do not show affinity for HSA, others are attracted and bind to HSA, resulting in an underestimation of the actual plasma metabolite concentration. After adding TSP, which is characterized by an even higher HSA affinity compared to the plasma metabolites, TSP will bind to HSA, resulting in the release of the bound metabolites. HSA, human serum albumin; TSP, trimethylsilyl-2,2,3,3-tetradeuteropropionic acid. Figure created with BioRender.com.

**Figure 4 ijms-25-04690-f004:**
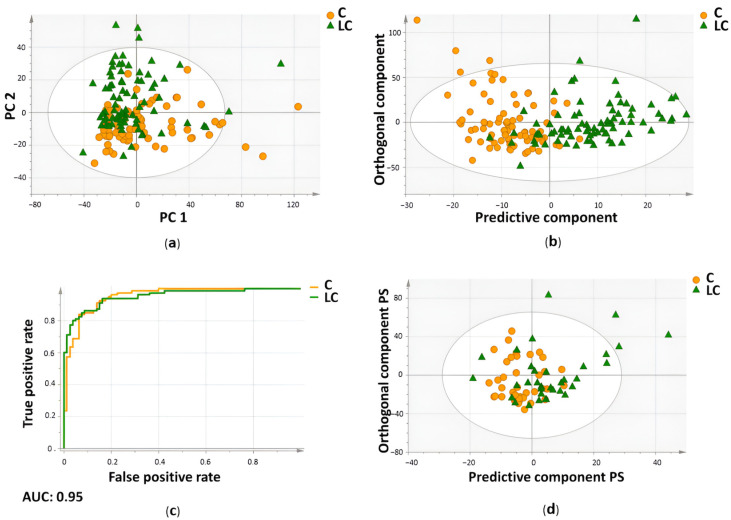
Differentiation between lung cancer patients and healthy controls based on metabolic profiling in plasma using a new methodology including TSP, as a human serum albumin (HSA)-binding competitor, and MA as an internal standard for metabolite quantification. (**a**) Unsupervised principal component analysis (PCA) shows a distinctive trend between the plasma metabolite profiles of the lung cancer patients (green triangles) and controls (orange circles). (**b**) Orthogonal partial least squares discriminant analysis (OPLS-DA) shows a clear distinction between the plasma metabolite profiles of the 80 lung cancer patients and 80 controls within the training cohort. The constructed model was characterized by a sensitivity and specificity of 85% and 93%, respectively. (**c**) The receiver operating characteristic (ROC) curve of the OPLS-training model confirmed the overall performance with an AUC of 0.95. (**d**) Validation of the constructed OPLS-DA model on an independent validation cohort resulted in a distinction between the plasma metabolite profiles of the 34 lung cancer patients and 38 controls with a sensitivity and specificity of 74% each. AUC, area under the curve; C, controls; LC, lung cancer patients; MA, maleic acid; PC, principal component; PS, predicted scores; TSP, trimethylsilyl-2,2,3,3-tetradeuteropropionic acid. Figure adapted from reference [20].

**Figure 5 ijms-25-04690-f005:**
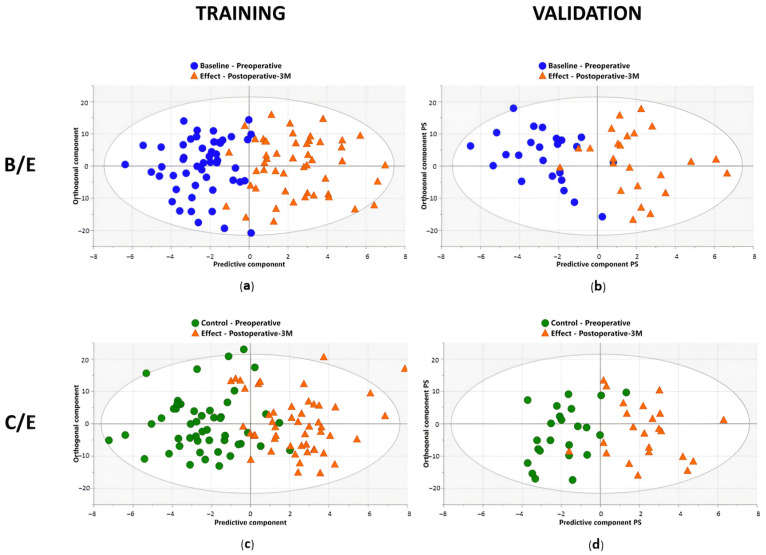
Differentiation between pre- and postoperative plasma samples based on metabolic profiling. The clear separation on the *x*-axis represents the metabolic variation between the pre- and postoperative groups, both when comparing baseline/effect (B/E) and control/effect (C/E). (**a**) Supervised orthogonal partial least squares discriminant analysis (OPLS-DA) shows an excellent distinction between the plasma metabolite profiles of preoperative (baseline: B, blue circles) and postoperative (effect: E (after 3 months), orange triangles) datasets within the training cohort consisting of 50 NSCLC patients. Sensitivity and specificity rates of 92% (92% of the preoperative samples correctly identified as preoperative) and 96% (96% of the postoperative samples correctly identified as postoperative), respectively, were reached. (**b**) Validation of the constructed OPLS-DA model in the validation cohort consisting of 24 NSCLC patients also resulted in a clear distinction between pre- and postoperative profiles with a sensitivity and specificity of 88% and 92%, respectively. (**c**) OPLS-DA model shows a great distinction between the preoperative (control; C; green circles) and postoperative datasets within the training cohort consisting of 50 NSCLC patients. Sensitivity and specificity rates of 88% and 90%, respectively, were reached. (**d**) Validation of the constructed OPLS-DA model in the validation cohort consisting of 23 NSCLC patients again resulted in a clear distinction between pre- and postoperative profiles with a sensitivity and specificity of 96% and 91%, respectively. PS, predictive scores. Figure adapted from reference [21].

**Figure 6 ijms-25-04690-f006:**
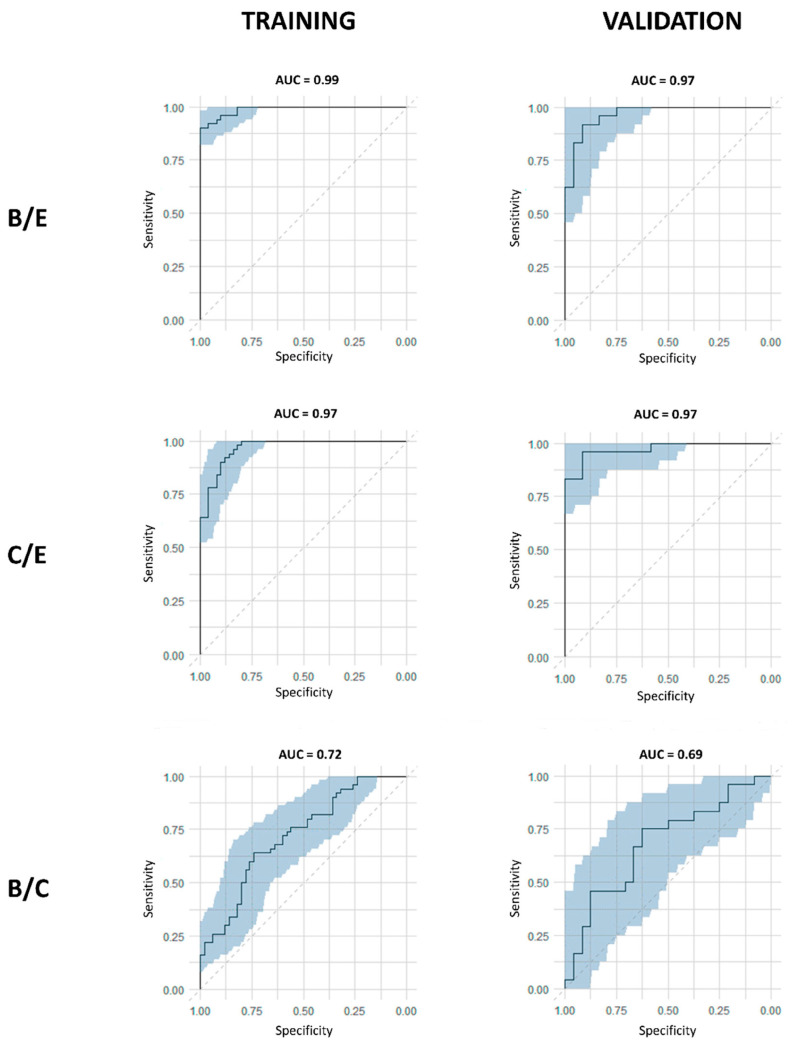
Overview of all the receiver operating curves (ROC), showing the area under the curve (AUC) values, in the three different comparisons, for both training and validation cohorts. Excellent AUC values were reached in both training and validation cohorts when comparing preoperative profiles (baseline: B, and control: C) with postoperative profiles (effect: E). The blue zone represents the 95% confidence interval of the obtained AUC values by internal validation via bootstrapping resampling. Figure reused from reference [21].

**Figure 7 ijms-25-04690-f007:**
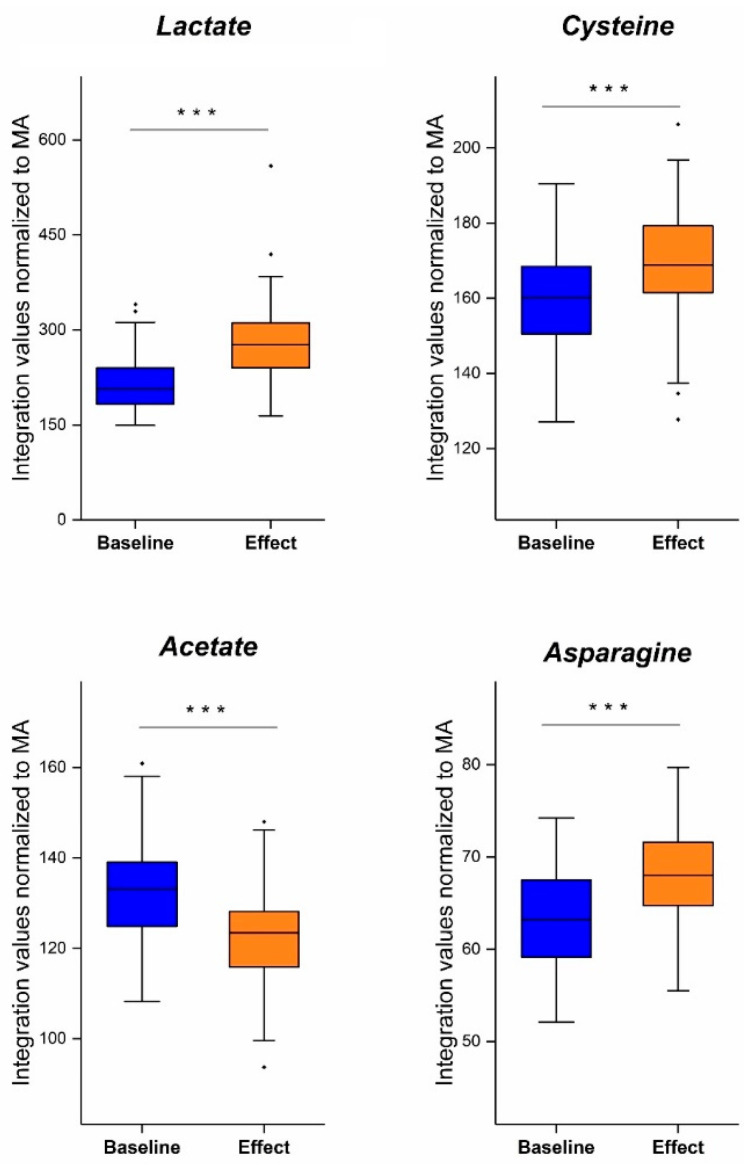
Integration values, normalized to MA, of lactate, cysteine, acetate, and asparagine in preoperative (baseline: blue) and postoperative datasets (effect: orange). Lactate, cysteine, and asparagine showed a significant increase in plasma concentrations after surgery. Acetate showed a significant decrease in plasma concentration after surgery. (*** *p* < 0.001) (· represents a datapoint outside the 1.5 IQR range). MA, maleic acid. Figure reused from reference [21].

**Figure 8 ijms-25-04690-f008:**
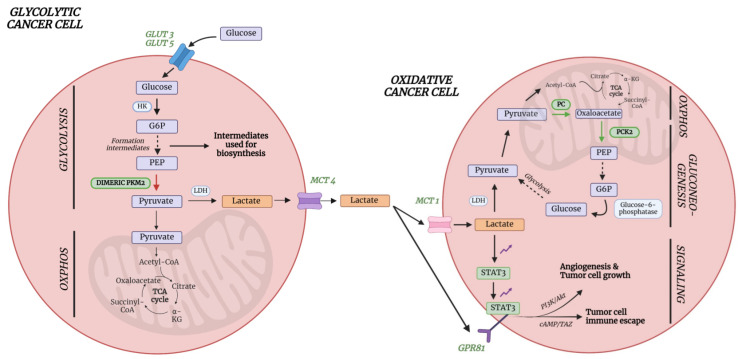
Metabolic symbiosis between glycolytic and oxidative lung cancer cells showing the lactate shuttle. Glycolytic cancer cells use aerobic glycolysis (aerobic fermentation) to obtain intermediates for biosynthesis, while oxidative phosphorylation is used to yield energy for further tumor growth. Nevertheless, glycolysis is mainly increased in the glycolytic cancer cells, while oxidative phosphorylation is decreased. As aerobic glycolysis is used, lactate is formed and released in the microenvironment by MCT4 transporters. Lactate can be taken up by oxidative cancer cells (“Lactate shuttle”) by the MCT1 transporters for gluconeogenesis. The glucose that is eventually formed will undergo glycolysis to stimulate biosynthesis via its intermediates and partially enter the TCA cycle after conversion to pyruvate to yield energy. Additionally, lactate will bind to the GRP81 receptor, which is upregulated by lactate itself via STAT3 in lung cancer cells, resulting in angiogenesis, tumor cell growth, and tumor cell immune escape. Enzymes, receptor labels, and arrows in green represent an upregulation, while red represents a downregulation. Akt, protein kinase B; α-KG, alpha-ketoglutarate; cAMP, cyclic adenosine monophosphate; GLUT3, glucose transporter 3; GLUT5, glucose transporter 5; GPR81, G-protein coupled receptor 81; G6P, glucose-6-phosphate; HK, hexokinase; LDH; lactate dehydrogenase; MCT1, monocarboxylate transporter 1; MCT4, monocarboxylate transporter 4; OXPHOS, oxidative phosphorylation; PC, pyruvate carboxylase; PEP, phosphoenolpyruvate; PCK2, phosphoenolpyruvate carboxykinase 2; PKM2, pyruvate kinase M2; PI3K, phosphoinositide 3-kinase; STAT3, signal transducer and activator of transcription 3; TAZ, transcriptional coactivator with PDZ-binding motif; TCA, tricarboxylic acid cycle. Figure created with BioRender.com.

**Table 1 ijms-25-04690-t001:** Characteristics of the study and statistical analysis.

Louis et al., 2016 [18]				
	Training Cohort		Validation Cohort	
	LC	C	LC	C
Number of subjects, n	233	226	98	89
Sensitivity (%)	78		71	
Specificity (%)	92		81	
R2X (Cum)	0.864		-	
R2Y (Cum)	0.477		-	
Q2 (Cum)	0.391		-	
AUC	0.88		0.84	

AUC, area under the curve; C, controls; LC, lung cancer patients; R2X (Cum), total explained variation in X within groups; R2Y (Cum), total explained variation in Y between groups; Q2 (Cum), predicted variation.

**Table 2 ijms-25-04690-t002:** Characteristics of the study and statistical analysis.

Louis et al., 2016 [19]				
	Training Cohort		Validation Cohort	
	LC	BC	LC	BC
Number of subjects, n	54	80	81	60
Sensitivity (%)	93		89	
Specificity (%)	99		82	
R2X (Cum)	0.82		-	
R2Y (Cum)	0.73		-	
Q2 (Cum)	0.63		-	
AUC	0.96		0.94	

AUC, area under the curve; C, controls; LC, lung cancer patients; R2X (Cum), total explained variation in X within groups; R2Y (Cum), total explained variation in Y between groups; Q2 (Cum), predicted variation.

**Table 3 ijms-25-04690-t003:** Characteristics of the study and statistical analysis.

Derveaux et al., 2021 [20]				
	Training Cohort		Validation Cohort	
	LC	C	LC	C
Number of subjects, n	80	80	34	38
Sensitivity (%)	85		74	
Specificity (%)	93		74	
R2X (Cum)	0.861		-	
R2Y (Cum)	0.581		-	
Q2 (Cum)	0.364		-	
AUC	0.95		-	

AUC, area under the curve; C, controls; LC, lung cancer patients; R2X (Cum), total explained variation in X within groups; R2Y (Cum), total explained variation in Y between groups; Q2 (Cum), predicted variation.

**Table 4 ijms-25-04690-t004:** Characteristics of the study and statistical analysis.

Derveaux et al., 2023 [21]					
	B/E		C/E		B/E
	OPLS-DA	OPLS-EP	OPLS-DA	OPLS-EP	OPLS-DA
TRAINING COHORT					
Number of subjects, n	50	50	50	50	50
Sensitivity (%)	92	*-*	88	-	74
Specificity (%)	96	-	90	-	62
R2X (Cum)	0.55	0.59	0.53	0.57	0.31
R2Y (Cum)	0.67	0.89	0.61	0.83	0.15
Q2 (Cum)	0.42	0.76	0.36	0.60	0.08
AUC	0.99	-	0.97	-	0.72
VALIDATION COHORT					
Number of subjects, n	24	24	23	23	23
Sensitivity (%)	88	-	96	-	74
Specificity (%)	92	-	91	-	43
AUC	0.97	-	0.97	-	0.69

Within the control group, one sample was identified as an outlier and excluded from further analyses. AUC, area under the curve; B, baseline (preoperative, before surgery); C, control (preoperative, at time of diagnosis); E, effect (postoperative, 3M after surgery); OPLS-DA, orthogonal partial least squares discriminant analysis; OPLS-EP, orthogonal partial least squares effect projections; R2X (Cum), total explained variation in X within groups; R2Y (Cum), total explained variation in Y between groups; Q2 (Cum), predicted variation.

## Data Availability

Not applicable.

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
