# Peer review of "Plasma Metabolite Profiling in the Search for Early-Stage Biomarkers for Lung Cancer: Some Important Breakthroughs"

_ijms, 2024, doi:10.3390/ijms25094690_

Round 1

Reviewer 1 Report

Comments and Suggestions for Authors

1.     Discuss the limitations and challenges associated with metabolomics and NMR metabolite profiling in lung cancer diagnosis. Addressing issues such as sample size, variability in metabolite levels, and standardization of analytical protocols would provide a more comprehensive understanding of the current state of the field.

2.     Provide a comprehensive overview of the current understanding of cancer metabolism and its implications for cancer biology and therapy. Highlighting key findings and controversies in the field would help readers appreciate the complexity of cancer metabolism and the challenges in targeting metabolic pathways for therapeutic purposes.

3.     Elaborate on the specific advantages of proton nuclear magnetic resonance (1H-NMR) spectroscopy over other techniques such as mass spectrometry for metabolic profiling. Providing a comparative analysis of different analytical methods would help readers understand the rationale behind choosing 1H-NMR for the study.

4.     Highlight specific breakthroughs or novel findings in the identification of early-stage biomarkers in the plasma of lung cancer patients using 1H-NMR spectroscopy. Emphasize the clinical significance and potential applications of these biomarkers in lung cancer screening, diagnosis, prediction, and follow-up monitoring.

5.     Provide a comprehensive discussion on the altered metabolic pathways associated with the identified biomarkers in the context of lung cancer. Elucidate the biological mechanisms underlying these metabolic changes and their implications for cancer development and progression.

6.     Conclude by summarizing the overall impact of the study findings and their implications for advancing early detection and management of lung cancer. Suggest potential future research directions or clinical applications based on the insights gained from the study.

Author Response

First of all, we would like to thank the reviewer for taking the time to review our manuscript and for the constructive comments. Please find our detailed responses below and our corresponding revisions/corrections marked in red in the re-submitted files.

2. Point-by-point response to Comments and Suggestions for Authors

Comment 1: Discuss the limitations and challenges associated with metabolomics and NMR metabolite profiling in lung cancer diagnosis. Addressing issues such as sample size, variability in metabolite levels, and standardization of analytical protocols would provide a more comprehensive understanding of the current state of the field.

Response 1: Thank you for pointing this out. In our discussion (7. Conclusion and future perspectives), we mention the main challenge for the breakthrough of NMR metabolomics and interpretation of changed metabolism, i.e. standardized analytical protocols with absolute metabolite quantification in plasma (line 876-877). Hereto, the addition of a HSA-binding competitor (TSP in our studies) and an internal standard (maleic acid in our studies) is of utmost importance. A HSA-binding competitor is needed since several metabolites bind to HSA and the amount of HSA can vary considerably among individuals and time, leading to HSA-dependent differences in measured plasma metabolite levels across different persons and different timepoints. An analytical internal standard (peak area/integration reference) is needed for peak area normalization instead of relative normalization towards the total area of all peaks. We extended this message in our review by adding a small part in the discussion, line 877-884)(in red).

Variability in plasma metabolite concentrations can, however, also be caused by other factors such as the disease stage. It is therefore of utmost importance that researchers establish clear inclusion criteria to understand/explain observed variation. Sample sizes are, indeed, also of great importance. When applying multivariate statistics, it is essential to use a sufficiently high sample size (number of patients and controls) to prevent overfitting of the data. We have added this message to the conclusion and future perspectives section of the paper, line 884-886 (in red).

Comment 2: Provide a comprehensive overview of the current understanding of cancer metabolism and its implications for cancer biology and therapy. Highlighting key findings and controversies in the field would help readers appreciate the complexity of cancer metabolism and the challenges in targeting metabolic pathways for therapeutic purposes.

Response 2: A comprehensive overview of the lung cancer metabolism is presented in section 6 of the manuscript “(Metabolic pathways involved in (lung) cancer”). Explanations of observed differences in metabolite profiles across the different studies are both explained in section 5 (“Discussion of breakthrough in NMR metabolomics research”) and section 6 (“Metabolic pathways involved in (lung) cancer”).

As mentioned above, one of the key findings of our research is that the addition of TSP as an HSA-binding competitor is crucial for several reasons. The addition of TSP (having a high affinity for HSA) will cause dissociation of HSA-bound metabolites such as lactate, acetate, phenylalanine, pyruvate... In the absence of TSP, the plasma concentrations of these metabolites will be significantly underestimated. In the study of Derveaux et al., in which TSP was used as an HSA-binding competitor, lactate, cysteine, and asparagine were found to be significantly decreased in the plasma of early-stage lung cancer patients, while acetate was found to be increased.

As mentioned in the review, controversies in plasma metabolite biomarkers can be explained through several issues:

  •          The absence of TSP as an HSA-binding competitor
    TSP should be added before accurate interpretation can occur (for the reason that was discussed above). As many studies did not use an HSA-binding competitor, metabolite variations arise.
  •          Different normalization method
    Many studies normalize the integration regions by dividing by the total integration area of the full 1H-NMR spectrum. Using this normalization method, plasma metabolites can only be determined relatively. Moreover, highly abundant plasma metabolites will dominate the total area and influence statistical analysis. For this reason, the use of an analytical internal standard is recommended. In the study of Derveaux et al. maleic acid (MA) was used as an internal standard. We have added this information to section 5, line 592-601 (in red).

  •          Different inclusion criteria
    As advanced stages show more metabolic alterations, the changes in plasma metabolite concentrations will differ from those in early-stages of the disease.
  •          Patient-to-patient variation
    In most metabolomics studies, a group of (lung) cancer patients is compared with a group of healthy controls or with patients diagnosed with a different type of cancer. While this type of study-set up provides a strong confirmation of the feasibility of differentiating two groups, which is needed for applying the metabolite profile in a diagnostic setting, individual follow-up of patients after treatment remains very challenging. As samples of the same individual were compared in the study of Derveaux et al., OPLS-EP was used instead of OPLS-DA, which considers the dependency of the sample pairs during the randomization process, resulting in even improved model interpretations.

Before addressing metabolic pathways for therapeutic purposes, a robust, quantitative method with an HSA-binding competitor and appropriate internal standard (as proposed by Derveaux et al.) is a must to avoid misleading interpretations. Afterwards, changes in metabolite levels should also be evaluated over time. Only then the disrupted metabolic pathways can be identified and therapeutic interventions can be considered. As we didn’t state the potential of metabolite biomarkers to target metabolic pathways for therapeutic purposes, we’ve added this to our conclusion (Section 7. “Conclusions and future perspectives”), line 907-909 (in red).

Comment 3: Elaborate on the specific advantages of proton nuclear magnetic resonance (1H-NMR) spectroscopy over other techniques such as mass spectrometry for metabolic profiling. Providing a comparative analysis of different analytical methods would help readers understand the rationale behind choosing 1H-NMR for the study.

Response 3: This was already mentioned in the introduction on line 99-102, but we improved it somewhat further (‘One of the main tools that is currently used for metabolic profiling, besides mass spectrometry, is proton nuclear magnetic resonance (1H-NMR) spectroscopy since the 1H nucleus is omnipresent in metabolites [16]. 1H-NMR has several advantages over mass spectrometry such as high reproducibility (>98%), short measuring time, and simple sample pre-treatment. As the detected peak areas correlate directly with the metabolite concentration, this technique also allows accurate quantitative analysis of substances even in complex mixtures such as plasma [16].’). We further added (parts in red in review) that these are specific advantages of NMR over mass spectrometry to make sure that this is clear for the reader. In this way, we hope that the rationale behind the choice for 1H-NMR is now more clear

Comment 4: Highlight specific breakthroughs or novel findings in the identification of early-stage biomarkers in the plasma of lung cancer patients using 1H-NMR spectroscopy. Emphasize the clinical significance and potential applications of these biomarkers in lung cancer screening, diagnosis, prediction, and follow-up monitoring.

Response 4: The goal of our research is to find early-stage biomarkers in the plasma of lung cancer patients to enable early disease detection in a quantitative manner (i.e. with TSP as a HSA-binding competitor), allowing identification and study of disrupted metabolic pathways. The breakthroughs of our research towards finding early-stage biomarkers in plasma of lung cancer patients using 1H-NMR, include:

-          The feasibility of NMR spectroscopy to use plasma metabolite biomarkers to diagnose early stage lung cancer (pre- and post-operative) and differentiate between cancer types,

-          The importance of the addition of TSP as a HSA-binding competitor to enable an accurate, quantitative analysis of the altered metabolism in lung cancer,

-          The potential of plasma biomarkers to monitor therapy/surgery response and to detect early disease recurrence in lung cance

In general, we aim to use 1H-NMR to provide support three stages of clinical decision making:

(1)    Combination of 1H-NMR and low-dose CT.
Before we can introduce 1H-NMR  as a clinical application to define the diagnostic status, a large clinical study should be set up where metabolite profiling of blood plasma via 1H-NMR and low-dose CT is combined for screening a high-risk population before clinical application as an initial screening tool.

(2)    The use of 1H-NMR to complement PET scans.
As explained in the introduction of the manuscript, PET scans often fail to differentiate between malignant and non-malignant PET-positive lesions, leading to a high number of false positives.

(3)    The use of 1H-NMR for disease prediction and monitoring.
The predictive and monitoring potentials of these metabolites are still under investigation. We are currently investigating the evolution of the metabolic plasma profiles at different postoperative time points (of the early-stage lung cancer patients included in the study of Derveaux et al). We are currently investigating the evolution of the metabolic plasma profiles at different postoperative time points (of the early-stage lung cancer patients included in the study of Derveaux et al). As mentioned in the study of Derveaux et al. (2023), 6 of the 80 eligible patients showed disease recurrence 6 months after surgery. These were initially excluded from the sub-study (but not from the clinical trial NCT03736993) that aimed to discriminate pre- and postoperative plasma metabolite profiles (i.e. early stage lung cancer vs. healthy metabolism model). Now, in a next step, this data can be used to evaluate whether the plasma metabolite profile allows to predict disease recurrence in these patients.

We have added this part to our conclusion (7. Conclusions and future perspectives), line 896-905 (in red).

To summarize, the proposed quantitative methodology definitely carries a great potential for future screening, diagnosis, prediction and therapy monitoring applications, but is not fully ready yet to be translated directly to the clinic. This was also added to the discussion of our review (7. Conclusions and future perspectives) from line 907-910 (parts in red).

Comment 5: Provide a comprehensive discussion on the altered metabolic pathways associated with the identified biomarkers in the context of lung cancer. Elucidate the biological mechanisms underlying these metabolic changes and their implications for cancer development and progression.

Response 5: A comprehensive discussion on the altered metabolic pathways associated with the identified biomarkers in the context of lung cancer is shown in section 6 (Metabolic pathways involved in (lung) cancer). Studies regarding the potential of these biomarkers to predict disease recurrence and progression (as mentioned above in response 4) are ongoing.

Comment 6: Conclude by summarizing the overall impact of the study findings and their implications for advancing early detection and management of lung cancer. Suggest potential future research directions or clinical applications based on the insights gained from the study.

Response 6: A summary of all important findings is provided at the end of each part of the review and in the final conclusions (Part 5 “Discussion of breakthroughs in NMR metabolomics research”). Future perspectives are shown in part 7 (“Conclusion and future perspectives”) in which we state that NMR-based metabolomics might be able to pave the way towards early cancer diagnosis via screening (of encumbered populations), therapy response, follow-up monitoring, and prediction of disease recurrence. Although the studies discussed already indicate the potential of the identified plasma metabolites as biomarkers (lactate, cysteine, asparagine, and acetate), In general, we aim to use 1H-NMR to provide support three stages of clinical decision making:

(1)    Combination of 1H-NMR and low-dose CT.
Before we can introduce 1H-NMR  as a clinical application to define the diagnostic status, a large clinical study should be set up where metabolite profiling of blood plasma via 1H-NMR and low-dose CT is combined for screening a high-risk population before clinical application as an initial screening tool.

(2)    The use of 1H-NMR to complement PET scans.
As explained in the introduction of the manuscript, PET scans often fail to differentiate between malignant and non-malignant PET-positive lesions, leading to a high number of false positives.

(3)    The use of 1H-NMR for disease prediction and monitoring.
The predictive and monitoring potentials of these metabolites are still under investigation. We are currently investigating the evolution of the metabolic plasma profiles at different postoperative time points (of the early-stage lung cancer patients included in the study of Derveaux et al). We are currently investigating the evolution of the metabolic plasma profiles at different postoperative time points (of the early-stage lung cancer patients included in the study of Derveaux et al). As mentioned in the study of Derveaux et al. (2023), 6 of the 80 eligible patients showed disease recurrence 6 months after surgery. These were initially excluded from the sub-study (but not from the clinical trial NCT03736993) that aimed to discriminate pre- and postoperative plasma metabolite profiles (i.e. early stage lung cancer vs. healthy metabolism model). Now, in a next step, this data can be used to evaluate whether the plasma metabolite profile allows to predict disease recurrence in these patients.

We have added this part to our conclusion (7. Conclusions and future perspectives), line 896-905 (in red).

Reviewer 2 Report

Comments and Suggestions for Authors

                The authors have provided an organized summary of several critical studies that have examined whether metabolite profiling by 1H-NMR can be used to detect lung cancer at an early stage. These studies have examined several distinctions, such as normals vs. lung cancer patients, breast cancer vs. lung cancer, and lung cancer before and after resection. Analysis in these studies have used a variety of approaches to avoid overfitting, and the ability of these distinction to be made has been quantified Studies have also shown that certain technical features improve performance: a more powerful magnetic field (600 MHz); addition of TSP to eliminate the effect of HSA binding; and use of malate as an internal standard. Finally, an extensive amount of knowledge about metabolism has been cited, to improve understanding of how lung cancer affects the metabolite profile.

                Reader comprehension of the review could be improved by addressing additional topics, although editing of the existing manuscript to reduce length would be required:

1) A brief presentation, if possible, of the technology of 1H-NMR that is being applied. At present, the manuscript only deals with the analysis of results, but how are those results generated? Most readers will have some understanding of the basic principles of electron spin and interrogation with radio frequency, to generate a spectrum based on the various different proton environments in the sample, but how are those data converted into measurements of different metabolites? The manuscript states that this is accomplished using 110 spectral regions based upon metabolite spiking, but illustration of a single example would be welcome. Also, how do we know that reference spectra are known for every metabolite in the sample?

2) A discussion of the point that in order to build models and assess performance, these studies have used samples from patients in whom the disease status is known, which carries a bias towards more advanced disease. However, what is desired is to know whether it would be possible to make the distinction, or at least determine the likelihood, of lung cancer being present at a very early stage. This problem also affects the study in which analysis was performed before and after resection. From a methods development standpoint, what studies need to be done to determine whether metabolite profiling by 1H-NMR is ready for clinical application in this most demanding setting? And how might it ultimately be applied, especially in comparison to the “liquid biopsy” to detect mutations in circulating tumor DNA? Should metabolite profiling by 1H-NMR be used as an initial screen, to decide which patients should receive additional tests?

3) Consideration of the “next patient” problem, related to #2. This term refers to a scenario such as the following. A study has been performed, with 200 training patients (100 normals, 100 cancer) and 100 validation patients (50 normals, 50 cancer), and a well-performing analytical model has been generated (see below). Using this model, are we ready to assign the next patient sample to a diagnostic status (cancer vs. normal)? If not, what more is needed to be able to do that?

4) A description of the analytical model, related to #3. This is just as important to the process as the methodology of #1, but equally opaque to the reader. How are the metabolite levels used to assign patient status? Is it a formula with weighted coefficients? Is it by application of orthogonal component vs. predictive component, which needs to be explained?

The sentence in Line 42 is nonsensical: “Early detection of lung cancer is however from utter importance to improve the survival rates [2].” Perhaps “however from utter” should be changed to “of utmost”.

Author Response

First of all, we would like to thank the reviewer for taking the time to review our manuscript and for the constructive comments. Please find our detailed responses below and our corresponding revisions/corrections marked in red in the resubmitted files.

2. Point-by-point response to Comments and Suggestions for Authors

Comment 1: A brief presentation, if possible, of the technology of 1H-NMR that is being applied. At present, the manuscript only deals with the analysis of results, but how are those results generated? Most readers will have some understanding of the basic principles of electron spin and interrogation with radio frequency, to generate a spectrum based on the various different proton environments in the sample, but how are those data converted into measurements of different metabolites? The manuscript states that this is accomplished using 110 spectral regions based upon metabolite spiking, but illustration of a single example would be welcome. Also, how do we know that reference spectra are known for every metabolite in the sample?

Response 1: We have tried to keep the review as understandable as possible for a broad public and have therefore omitted some of the technical details. However, we agree that this should be available to people who are interested. We have added this additional information to the Appendix, starting from line 939 (in red), in which we also included a figure (Figure A1), in which an example of metabolite identification is shown. We also added some extra information about the spectral regions and how they represent the variables used for further statistical analysis on line 161-163 and line 978-980. Readers that are interested in more technical details can find extra information via the references of the related papers that are now provided to each study design (‘More technical details can be found in Appendix A or in the paper of …’). Additionally, we have also added additional references for more information about the OPLS-DA and OPLS-EP on line 195-196 and 521-523 (in red). These publications are shown below:

-          Eriksson L., J.E., Kettaneh-World N., Trygg J., Wikström C., Wold S. Multi- and Megavariate Data Analysis Basic Principles and Applications; Umetrics Academy: Umeå: Sweden, 2013. (OPLS-DA information)

-          Jonsson, P.; Wuolikainen, A.; Thysell, E.; Chorell, E.; Stattin, P.; Wikström, P.; Antti, H. Constrained randomization and multivariate effect projections improve information extraction and biomarker pattern discovery in metabolomics studies involving dependent samples. Metabolomics 2015, 11, 1667-1678, doi:10.1007/s11306-015-0818-3. (OPLS-EP information)

Comment 2: A discussion of the point that in order to build models and assess performance, these studies have used samples from patients in whom the disease status is known, which carries a bias towards more advanced disease. However, what is desired is to know whether it would be possible to make the distinction, or at least determine the likelihood, of lung cancer being present at a very early stage. This problem also affects the study in which analysis was performed before and after resection. From a methods development standpoint, what studies need to be done to determine whether metabolite profiling by 1H-NMR is ready for clinical application in this most demanding setting? And how might it ultimately be applied, especially in comparison to the “liquid biopsy” to detect mutations in circulating tumor DNA? Should metabolite profiling by 1H-NMR be used as an initial screen, to decide which patients should receive additional tests?

Response 2: Thank you for this interesting and important remark. We are aware that, for screening with 1H-NMR (or other methods like low dose CT), the number of early stage lung cancer patients is very (too) limited in the general population. Before we can introduce 1H-NMR as an initial screening test, a large clinical study should be set up where metabolite profiling of blood plasma via 1H-NMR and low-dose CT is combined for screening a high-risk population.

Our other aims with 1H-NMR are stated in response 3 below.

Comment 3: Consideration of the “next patient” problem, related to #2. This term refers to a scenario such as the following. A study has been performed, with 200 training patients (100 normals, 100 cancer) and 100 validation patients (50 normals, 50 cancer), and a well-performing analytical model has been generated (see below). Using this model, are we ready to assign the next patient sample to a diagnostic status (cancer vs. normal)? If not, what more is needed to be able to do that?

Response 3: The proposed technique is not fully ready yet to be translated directly to the clinic. In general, we aim to use 1H-NMR to provide support three stages of clinical decision making:

(1)    Combination of 1H-NMR and low-dose CT.
Before we can introduce 1H-NMR  as a clinical application to define the diagnostic status, a large clinical study should be set up where metabolite profiling of blood plasma via 1H-NMR and low-dose CT is combined for screening a high-risk population before clinical application as an initial screening tool.

(2)    The use of 1H-NMR to complement PET scans.
As explained in the introduction of the manuscript, PET scans often fail to differentiate between malignant and non-malignant PET-positive lesions, leading to a high number of false positives.

(3)    The use of 1H-NMR for predictive and monitoring.
The predictive and monitoring potentials of these metabolites are still under investigation. We are currently investigating the evolution of the metabolic plasma profiles at different postoperative time points (of the early-stage lung cancer patients included in the study of Derveaux et al). We are currently investigating the evolution of the metabolic plasma profiles at different postoperative time points (of the early-stage lung cancer patients included in the study of Derveaux et al). As mentioned in the study of Derveaux et al. (2023), 6 of the 80 eligible patients showed disease recurrence 6 months after surgery. These were initially excluded from the sub-study (but not from the clinical trial NCT03736993) that aimed to discriminate pre- and postoperative plasma metabolite profiles (i.e. early stage lung cancer vs. healthy metabolism model). Now, in a next step, this data can be used to evaluate whether the plasma metabolite profile allows to predict disease recurrence in these patients.

We have added this information to our conclusion (7. Conclusion and Future perspectives) on line 896-905 (in red).

To summarize, the proposed quantitative methodology definitely carries a great potential for future screening, diagnosis, prediction and therapy monitoring applications, but is not fully ready yet to be translated directly to the clinic. We also added this to our conclusion on line 907-910 (in red).

Comment 4: A description of the analytical model, related to #3. This is just as important to the process as the methodology of #1, but equally opaque to the reader. How are the metabolite levels used to assign patient status? Is it a formula with weighted coefficients? Is it by application of orthogonal component vs. predictive component, which needs to be explained?

Response 4:  To make sure that sufficient technical details can be found for those who are interested, we have added an Appendix and references of the related papers to each study design (‘More technical details can be found in Appendix A or in the paper of …’). Additionally, we have also added additional references for more information about the OPLS-DA and OPLS-EP on line 195-196 and 521-523 (in red). These publications are shown below:

-          Eriksson L., J.E., Kettaneh-World N., Trygg J., Wikström C., Wold S. Multi- and Megavariate Data Analysis Basic Principles and Applications; Umetrics Academy: Umeå: Sweden, 2013. (OPLS-DA information)

-          Jonsson, P.; Wuolikainen, A.; Thysell, E.; Chorell, E.; Stattin, P.; Wikström, P.; Antti, H. Constrained randomization and multivariate effect projections improve information extraction and biomarker pattern discovery in metabolomics studies involving dependent samples. Metabolomics 2015, 11, 1667-1678, doi:10.1007/s11306-015-0818-3. (OPLS-EP information)

We have also added an extra line to provide more information about the integration regions on line 161-163 and 995-997 (in red): ‘The integration regions are the variables for the multivariate statistics and the value of each variable is determined by the area under the peak of the integrations region (and therefore directly correlated with the concentration of a specific metabolite).’

We’ve also added a short line about the analysis that was used to identify the contributing metabolites on line 190-191 (in red): ‘The variables responsible for the observed differentiation can be analyzed using the variable of importance projection (VIP) analysis’.
